# The Translational Impact of Plant-Derived Xeno-miRNA miR-168 in Gastrointestinal Cancers and Preneoplastic Conditions

**DOI:** 10.3390/diagnostics13162701

**Published:** 2023-08-18

**Authors:** Jastin Link, Cosima Thon, Vytenis Petkevicius, Ruta Steponaitiene, Peter Malfertheiner, Juozas Kupcinskas, Alexander Link

**Affiliations:** 1Department of Gastroenterology, Hepatology and Infectious Diseases, Otto-von-Guericke University, 39120 Magdeburg, Germanycosima.thon@med.ovgu.de (C.T.); peter.malfertheiner@med.ovgu.de (P.M.); 2Department of Gastroenterology and Institute for Digestive Research, Lithuanian University of Health Sciences, 44307 Kaunas, Lithuania; vytenis.petkevicius@lsmuni.lt (V.P.); ruta.steponaitiene@lsmuni.lt (R.S.); juozas.kupcinskas@lsmuni.lt (J.K.); 3Medical Department II, University Hospital, Ludwig-Maximilians-Universität, 80539 Munich, Germany

**Keywords:** microRNA, diet, xeno-miRNAs, gastric cancer, gastritis, miR-168

## Abstract

Introduction: Diet is one of the most important factors contributing to the multistep process of carcinogenesis. The clinical relevance of exogenous food-derived xeno-microRNAs (miRNAs) in human diseases is poorly understood. In this study, we aimed to evaluate the potential clinical relevance of the xeno-miRNA miR-168 in the gastric mucosa along the preneoplastic conditions and gastric carcinogenesis. Methods: For a systematic analysis, we included stomach tissues from patients with different pathologies, including normal mucosa (N), chronic non-atrophic (CNAG) and atrophic gastritis (CAG) and intestinal metaplasia (IM) (n = 72), matched non-tumorous (NT) and tumorous (T) gastric cancer (GC) tissues (n = 81), matched colorectal cancer (CRC) tissues (n = 40), and colon mucosa and faeces from controls and IBD patients. Results: miR-168 was reproducibly detectable in all samples studied, with the highest levels in the proximal upper GI and in non-tumorous compared to tumorous tissues in both GC and CRC. There was no difference related to *H. pylori* positivity or inflammation grade, while higher miR-168 levels were observed in patients with moderate or severe AG/IM or OLGIM3/4. Survival analysis showed only a small, non-significant trend towards worse overall survival for patients with the highest to lowest miR-168 levels, while no differences were related to Lauren‘s classification. Conclusions: Food-derived xeno miRNAs are reproducibly detectable in the gastric and colonic mucosa. Although the clinically relevant function remains to be elucidated, higher levels of miR-168 in patients with moderate and severe IM merit further investigation.

## 1. Introduction

MicroRNAs (miRNAs) are short, non-coding, highly conserved, single-stranded, functionally active RNA molecules involved in post-transcriptional gene silencing [1]. It is therefore not surprising that miRNAs are involved in various, if not all, cellular functions, e.g., as regulators of epigenetic modifications of genes, relevant in immune response, cell proliferation, apoptosis, etc. [2]. Different types of miRNAs are dysregulated in cancer, e.g., miR-21, or are involved in the regulation of inflammation, e.g., miR-155 is involved in inflammatory processes [3,4,5]. Accumulating data in recent years clearly support the functional role of miRNAs in various cancers, including gastric cancer (GC) and colorectal cancer (CRC) [6,7]. Furthermore, miRNAs are well preserved in tissues and biological samples such as blood, faeces or ascites, and are considered to be potential diagnostic and prognostic biomarkers, although several challenges remain in their evaluation [8,9,10,11].

It is now well-known that miRNAs are not only found in mammals and other animals, but also in plants [12,13]. Systematic analysis has revealed a remarkable number of plant miRNAs, of which miR-168-5p (miR-168) may be one of the most abundant. Those miRNAs that have been identified in other species or even across kingdoms are called xeno-miRNAs [14]. While the translocation, such as via breastfeeding, can be well explained biologically, cross-kingdom translocation, for example from plant miRNAs to humans, remains poorly understood. In one of the first papers, Zhang et al. showed that exogenous plant-derived miR-168 was not only identified in liver tissue, but could also cause downregulation of LDLRAP1 (human low-density lipoprotein receptor adaptor protein 1) in mouse livers, with a subsequent increase in serum LDL [15]. Since miR-168 has been shown to reduce LDLRAP1 levels in liver tissue, it is suggested that miR-168 is able to pass through the GI tract. While studies have reported the detection of miR-168 in sera, others have failed to detect the presence of xeno-miR-168 in circulation [15,16,17,18,19].

In our previous study, we convincingly demonstrated that animal and plant miRNAs are present in virtually all foods, regardless of processing [16]. Since dietary miRNAs such as miR-21-5p or miR-155p-5p are highly conserved, it is impossible to estimate the effect of dietary miRNAs in humans. Therefore, we used miR-168 as a model to more precisely evaluate the effects of dietary miRNAs on changes in humans, for example in faeces and blood serum. Healthy volunteers showed an increase in faecal miR-168 following a vegetarian diet, suggesting a causal relationship between diet and the detection of plant miRNAs in human samples. While miR-168 was not detectable in blood samples, we reported the reproducible detectability of plant-derived miR-168 not only in gastric and colonic mucosa, but also in ascites from patients with cirrhosis [16]. Furthermore, in our preliminary data, we were able to detect xeno-miRNAs not only in non-tumorous colon tissue, but also in tumorous colon cancer; however, the biological and translational relevance of xeno-miRNAs in the gastrointestinal tract, and particularly in the stomach, is still unknown.

Gastric cancer (GC) is not only one of the most common cancers worldwide, but also one of the best studied cancer models [20,21,22]. The gastric mucosa is also the first barrier to interact intensively with exogenous factors, including food. Although *Helicobacter pylori* (*H. pylori*) is considered to be one of the most critical risk factors [23], other factors, including diet, may be responsible for cancer progression, particularly in the progression from chronic gastritis to the preneoplastic conditions atrophic gastritis (AG) and intestinal metaplasia (IM), or in the later stages of progression to dysplasia and invasive cancer along Correa’s cascade of gastric carcinogenesis. The role of xeno-miRNAs in gastric carcinogenesis has not yet been investigated.

In this study, we systematically investigated the potential clinical relevance of the xeno-miRNA miR-168 in gastric carcinogenesis. First, we performed a quantitative analysis of miR-168 in non-tumour (NT-GC) and tumour (T-GC) gastric and CRC tissues and its impact on disease phenotype, including overall survival. Second, we investigated the detectability of miR-168 in preneoplastic conditions and in relation to *H. pylori* infection.

## 2. Materials and Methods

### 2.1. Ethics

This prospective study was conducted in accordance with the World Medical Association Declaration of Helsinki—Ethical Principles for Medical Research Involving Human Subjects. All tissue samples were collected in two clinical departments: the Department of Gastroenterology and Surgery, Hospital of Lithuanian University of Health Sciences (Kaunas, Lithuania) protocol no. 2/2008, and the Department of Gastroenterology, Infectiology and Hepatology, Otto-von-Guericke University Magdeburg (approval no. 80/11 and 65/19). Written informed consent was obtained from patients before enrolment.

### 2.2. Surgical Tissue Specimens

Tissue samples from GC and CRC patients were taken during surgery. A total of 81 pairs of tumour (T-GC) and adjacent non-tumour gastric tissues (NT-GC) were obtained from the stomach. The Lauren classification and the International Classification of Diseases for Oncology were used for histological classification of GC patients. A subset of tumour samples (n = 28, T-CRC) and adjacent non-tumour tissues (n = 26, NT-CRC) were also used to assess the impact in CRC patients. The detailed characteristics of the cohorts and the description of storage and processing have been described previously [24,25,26,27].

### 2.3. Biopsies from Gastric and Colon Mucosa

Gastric antrum biopsies were obtained during routine upper GI endoscopy after overnight fasting. We included 30 control patients with normal gastric mucosa (N), 20 patients with chronic gastritis (CG) and 22 patients with AG/IM. All samples were histologically characterized according to the updated Sydney classification as previously described [28]. *H. pylori* status was assessed by rapid urease test, serology, histology and microbiological culture. *H. pylori* positivity was defined on the basis of direct identification of *H. pylori* in histology and/or microbiology. The status of *H. pylori* infection was determined by rapid urease test, serology, histology and microbiology. Patients were defined as *H. pylori* positive if microbiology was positive and/or histology was positive. For *H. pylori* culture, gastric samples were stored in 0.9% by volume isotonic sodium chloride solution (Berlin-Chemie AG, Berlin, Germany). Biopsies were cultured on Columbia agar-based medium with and without antibiotic supplementation (10 mg/mL vancomycin, 1 mg/mL nystatin and 5 mg/mL trimethropin). Plates were incubated under microaerophilic conditions (37 °C, 5% CO_2_) for a maximum of 10 days and bacterial growth was checked every 2–3 days. *H. pylori* was identified by typical morphology on Gram-negative staining and positive urease, oxidase and catalase tests, as previously described [29].

Colon biopsies and faecal samples were obtained from the control cohort and patients with ulcerative colitis (UC) and Crohn’s disease (CD), n = 15 for each. All patients with UC and CD had clinically and histologically confirmed disease. Colon biopsies were obtained during sigmoidoscopy or colonoscopy after enema or standard bowel preparation. Biopsies were taken from the distal sigmoid colon. Faecal samples were collected during routine examinations and handled as previously described [30].

### 2.4. RNA and DNA Isolation

For RNA isolation from biopsy specimens, we used the Qiagen miRNeasy Mini Kit (Qiagen, Hilden, Germany) according to the manufacturer’s instructions with minimal modifications as previously described [16]. Briefly, frozen biopsy samples were first lysed in 900 μL QIAzol Lysis Reagent (Qiagen, Hilden, Germany) and then homogenised using the TissueRuptor. After homogenisation, the RNA was precipitated with 180 μL of chloroform, and 1.5 volumes of 100% ethanol were used to mix the upper aqueous phase. After washing, the extracted RNA was eluted.

RNA and DNA isolation from surgical material was performed as previously described [25]. Briefly, total RNA was extracted using the RNeasy Plus Universal MiniKit (Qiagen, Hilden, Germany), where 20–30 mg of frozen samples were lysed using QIAzol Lysis Reagent. After homogenisation, the RNA was precipitated with chloroform and the upper aqueous phase was mixed with ethanol and eluted using a spin column after washing. DNA was isolated using QIAzol Lysis Reagent and chloroform in the interphase.

RNA isolation from faecal material was performed in a similar manner to biopsies, homogenising similar amounts of faeces and adding exogenous cel-miR-39 for internal normalisation, as previously reported [16]. After extraction, samples were further analysed spectrophotometrically using a biophotometer (Eppendorf, Germany) and samples were stored at −80 °C until further analysis.

### 2.5. miR-168 Expression Analysis

To quantify miR-168 expression, we used the TaqMan miRNA assay ID 007594_mat (Applied Biosystems, Foster City, CA, USA) as previously described [16]. To quantify miRNA expression using a TaqMan assay, approximately 20 ng of RNA was reverse transcribed according to the manufacturer’s protocol, and quantitative real-time PCR analyses were performed using the BioRad CFX Cycler System (BioRad, Hercules, CA, USA). Normalisation was performed according to the 2^ΔCT^ methods using the small nuclear RNA RNU6b gene or to cel-miR-39 in faecal specimens (TaqManAssay ID: 000200) [30].

### 2.6. Correlation with Molecular Characteristics

Purified genomic DNA was bisulfite modified using the Cells-to-CpGtu Bisulfite Conversion Kit (Life Technologies, Carlsbad, CA, USA) as previously described, and miR-137 CpG island and long interspersed nucleotide element-1 (LINE-1) methylation analyses were performed [24,25]. To evaluate for further potential interaction, previously available data for HOTAIR expression and data to *Fusobacterium nucleatum* (*F. nucleatum* or FNUC) was used [26,27].

### 2.7. Statistical Analysis

GraphPad Prism 6.0 (San Diego, CA, USA) was used for statistical analysis. Since our miRNA data were not normally distributed, a non-parametric test was used. For qualitative analysis, the χ^2^ (chi square) test was performed. The Wilcoxon test was used for paired variables and the Mann–Whitney U test for unpaired variables. Kruskal–Wallis analyses of variance were used to analyse statistical significance for multiple group comparisons, with the appropriate Dunn’s multiple comparison test for post hoc analyses. Spearman’s test was applied for correlation analysis. Overall survival data were available for GC and CRC samples up to 2500 days. Data sets for patients with less than 30 days post-surgery were excluded to avoid any surgery-related bias. Kaplan–Meier survival analyses were performed for patients with available survival data, and the log-rank (Mantel–Cox) test was used to compare survival curves. Two-sided *p* values of <0.05 were considered statistically significant in applied tests.

## 3. Results

### 3.1. miR-168 in Nonneoplastic Mucosa

To investigate the difference in miR-168 along the GI tract, we first compared the levels in stomach and colon mucosa. As shown in Figure 1A, the level of miR-168 was higher in the stomach (both N and NT-GC) compared to NT-CRC. Interestingly, the comparison of miR-168 in preneoplastic conditions showed the highest level in NT-GC (*p* < 0.01), while the difference between N, CNAG and AG/IM groups did not reach a statistically significant difference. Next, we asked whether *H. pylori* could affect miR-168 levels in the gastric mucosa, but we observed no significant difference in miR-168 levels between subjects with and without *H. pylori* infection (N and GC combined) (Figure 1C). The extent of inflammation, measured histologically by PMN or lymphocyte infiltration, showed no difference (Figure 1D,E). Next, we asked whether the presence of AG or IM could be associated with a difference in miR-168, but this was not the case (Figure 1F). However, when considering the severity, patients with AG/IM grade 2 and 3 as well as OLGIM3–4 showed significantly high miR-168 levels compared to normal mucosa or patients with grade 1 or OLGIM1/2 (*p* < 0.01). (Figure 1G,H).

### 3.2. miR-168 in Neoplastic Mucosa

When comparing NT-GC and T-GC samples, there was a lower level of miR-168 in T-GC compared to NT-GC (*p* < 0.001) (Figure 2A). This was also the case using paired data visualization (Figure 2B). Correlation analysis revealed an overall significant correlation between the level in NT-GC and T-GC, suggesting patient-specific differences (*p* = 0.0309). The clinicopathological characteristics of GC patients in relation to miR-168 levels are shown in Table 1. Having shown the decreasing levels of miR-168 towards the distal parts of the GI tract, we question whether miR-168 might have different concentrations in T-GC depending on the primary region of the tumour. Interestingly, although as expected, there was a gradual decrease in miR-168 levels from the proximal to the distal part of the stomach, assuming that degradation or dwell time could possibly be responsible for the decrease (*p* < 0.001, Figure 2D).

### 3.3. miR-168 and Overall Survival

Having shown the differences in levels, we next asked whether miR-168 might be associated with disease prognosis, and therefore performed an overall survival analysis of different miR-168 levels in T-GC (Figure 3). First, we examined the overall survival of the entire cohort in non-tumour tissues using the median as a cut-off (Figure 3A). As shown in Figure 3, there was no significant difference between patients with high or low levels of miR-168 in NT-GC, regardless of Lauren type. Analysis of T-GC samples showed a slightly better prognosis for tumours with high miR-168 levels, especially in diffuse type cancers, but the difference did not reach significance (Figure 3B–D). Due to the distribution of miR-168 within the T-GC cohort, we observed a certain cluster of very high miR-168 levels in the T-GC samples (Figure 3A). Interestingly, Kaplan–Meyer curves showed a trend for a better overall survival of subjects with high levels of miR-168 in the T-GC patients, suggesting a potential functional impact, although an indirect association between residual dietary intake and higher miR-168 levels may also be responsible for the difference (Figure 3B).

### 3.4. miR-168 and Other Molecular Features

Several molecular features have previously been shown to be associated with the clinical phenotype of GC and have been linked to the prognosis of GC patients. For example, LINE-1 DNA methylation is a common event in carcinogenesis and is a surrogate for global genomic DNA hypomethylation [24]. miR-137 promoter methylation has been linked to the stepwise cascade of carcinogenesis in both GC and CRC [25]. LncRNA HOTAIR is a relatively new prognostic marker in GC patients [26]. The microbiome (besides *H. pylori*) is gaining increasing attention in the assessment of GC pathogenesis, and is likely to have an impact on the prognosis of patients with GC [31]. In particular, *F. nucleatum* has been associated with prognosis in GC patients [27]. Assuming that miR-168 levels may be related to specific molecular alterations, we next correlated miR-168 levels with available data on molecular characterization of the samples, including LINE-1 DNA methylation and miR-137 promoter DNA methylation levels, HOTAIR expression and *F. nucleatum* positivity (Figure 4A–D). While no difference in miR-168 was observed between tumours positive or negative for HOTAIR or *F. nucleatum*, we observed a positive correlation between LINE-1 methylation and miR-168 levels (*p* < 0.0068) (Figure 4A). A positive correlation was also observed between miR-137 promoter methylation and miR-168 levels (*p* = 0.0107). Whether this association could be related to a functional difference remains to be clarified.

### 3.5. miR-168 in CRC

Having analysed the impact of miR-168 on prognosis in GC patients, we ask whether miR-168 might be associated with specific tumour behaviour in CRC patients. Similar to GC, we observed a significantly higher miR-168 level in NT-CRC compared to T-CRC (*p* < 0.001, Figure 5A,B). Correlation analysis also showed a significant coherence in levels between T-CRC and NT-CRC (*p* < 0.0001) (Figure 5C). However, overall survival analysis showed no significant difference between subjects with high or low miR-168 levels in T-CRC (Figure 5D). To explore whether chronic inflammation may be associated with the higher or lower levels of miR-168, we extended our analysis to the tissue and faecal analysis of the IBD patients. As shown in Figure 6, there was a slight increase in miR-168 in the colonic mucosa of Crohn’s disease patients but, despite the trend, the difference was not significant.

## 4. Discussion

Systematic analysis over the past year has shown that miRNAs not only exert their function in the cell in which they are produced, but also, like hormones, in other regions after internalisation, for example the exosomes. While this seems to be the case for endogenous miRNA biogenesis, the biogenesis of allogenic miRNAs is still poorly understood. Even less is known about the cross-kingdom interactions, for example, between miRNAs from plants and humans. Previous studies have reported the detectability of plant miR-168 in humans, but the role and function of these miRNAs are insufficiently understood [16].

In this work, we performed a systematic analysis of miR-168 in gastric mucosa of patients with different types of gastritis and along Correa’s cascade of carcinogenesis and the impact on prognosis in GC and CRC patients. Our data suggest that miR-168 is present at higher levels in the stomach compared to the colon, and that T-GC and T-CRC have lower miR-168 levels compared to non-tumorous tissues. While *H. pylori* infection and general gastric inflammation had no effect on miR-168 levels, the severity of AG/IM was associated with increased miR-168 levels.

The distribution of xeno-miRNAs in the GI tract has not been well studied [15]. Assuming that the majority of the xeno-miRNAs pass through the main oral–anal route, it can be speculated that the highest concentration can be expected in the proximal and the lowest in the distal part of the GI tract. A comparison of stomach and colon samples showed higher levels of miR-168 in the stomach compared to the colon. Unfortunately, we did not analyse the small intestine samples and are therefore unable to provide any information. If the hypothesis is correct, one would expect higher miR-168 levels in the proximal tumour compared to the distal tumour of the stomach. The data showed that cardia tumours had significantly higher levels of miR-168 with a stepwise decrease from corpus to antrum, supporting the hypothesis. Furthermore, although in previous studies there was only a limited effect of various factors such as heating and RNAse treatment on the stability of miR-168, we still believe that gastric acidity could affect miR-168 levels in the stomach. The importance of the stomach in the uptake of exogenous miRNA has recently been highlighted. Chen et al. linked SIDT1, which is involved in intercellular miRNA transport and uptake, to miRNA absorption in the stomach [32]. Interestingly, the highly acidic environment was critical for the SIDT1-dependent absorption of miRNA. Furthermore, the authors showed that dietary absorption of miRNAs (plant-derived miR-2911) prevented the liver fibrosis that occurred in SIDT1-deficient mice. These data further highlight the critical role of the stomach in xeno-miRNA biogenesis.

In our previous publication, we reported a higher level of miR-168 in NT-CRC compared to T-CRC. As the data were limited by the small sample size, we expanded the dataset. Confirming our previous findings, we show that both GC and CRC tissues have lower miR-168 levels compared to adjacent tissues. Although the exact mechanism is not clear, one can speculate that potential internalization of exogenous miRNAs might be more difficult in tumour tissues. As the GC and CRC samples were obtained during surgery, we believe that the potential bias related to the endoscopic procedure can be excluded and that the miR-168 obtained deep in the tumour is of potential functional relevance.

Nevertheless, evaluation of the role of miR-168 in the biology of GC and CRC revealed that only patients with very high levels of miR-168 had a slightly better prognosis, but the data are limited by the dataset and further analysis is needed. Whether this borderline association is due to a tumour-independent dietary behaviour or a functional role of miR-168 remains to be clarified. GC carcinogenesis is a stepwise process involving multiple aetiologies and molecular pathways at genetic and epigenetic levels. We took advantage of the molecular characterisation of the samples and evaluated the potential interaction between miR-168 and DNA methylation, HOTAIR positivity and *F. nucleatum* level [24,25,26,27]. Nevertheless, only higher miR-168 correlated positively with miR-137 and LINE-1 methylation, suggesting a potential biological role. Although we have addressed some of the key characterisations in relation to GC, including the assessment of Lauren’s classification, this may only provide a partial view and many more molecular pathways need to be considered. For example, mismatch repair deficiency (MMR) is one of the key pathways leading to hypermutation and microsatellite instability (MSI) that can trigger carcinogenesis, and is present in up to 15% of GI cancers [33,34]. Epstein–Barr virus (EBV) is also known to be a pathogenic factor in a subset of GCs with a unique methylation profile [35]. The frequency of EBV in GC is well estimated and recent data suggest a functional role of EBV in host cell signalling and modulation of immune interactions [36,37]. The frequency of EBV in GC is estimated to be up to 5–10% depending on GC localisation [38]; the estimated positivity in our cohort would reach up to 5 samples, which may be too low to provide an adequate assessment of translational interaction and was therefore not evaluated in this work.

The discussion about the relevance of xeno-miRNAs for the physiology of an organism is still ongoing, but the data from our study again reproducibly show that xeno-miR-168 can be detected in biological samples [16]. miR-168 was not only detected in gastric tumour and non-tumour tissues, but also in different parts of the colon and in colorectal cancer. Further exploration of the pathophysiological relevance will provide potential opportunities in disease prevention, diagnosis or even treatment. However, despite the fact that the data provided in this work provide an additional puzzle to understanding the role of xeno-miRNAs in humans, there are several limitations. Firstly, the data provide only associative evidence and further mechanistic studies are needed. In this case, in vitro 3D cell culture of organoids may provide a deeper insight into the physiological interaction. In addition, animal experiments could potentially provide the knowledge that is difficult to obtain due to heterogeneity and fill the missing gap. Finally, we focused on miR-168 as one of the most abundant and highly concentrated miRNAs in the plant diet, but other miRNAs may be of great physiological relevance and deserve attention. In particular, a growing body of data links plant-derived miRNAs to the gut microbiome, suggesting that cross-kingdom interactions may also be indirect [39].

Our hypothesis of a functional role for xeno-miRNAs is supported by recent in vitro data supporting a functional role for exogenous miRNAs in cellular homeostasis. For example, shistosomal miRNAs was shown to promote hepatic fibrosis by targeting the trans-formin growth factor-β signalling pathway [40]. In another study, miR-166a from *Lycium barbarum* was investigated in vitro in a kidney cancer model highlighting the antitumor properties [41]. Soybean-derived miRNAs inhibited proliferation and induced apoptosis in a human cancer cell line in vitro [42] or in mice [43].

In conclusion, exogenous plant-derived miRNA miR-168 was reproducibly detectable in human gastric, colonic and faecal mucosa. Decreasing concentration from the proximal to the distal part of the GI tract might be related to the ongoing degradation of internalization of xeno-miRNAs in the GI tract. Lower miRNA expression in tumour tissues suggest the potential tumour-specific biology and changes in the internalization of xeno-miRNAs. A trend towards a better prognosis of GC patients with very high levels of miR-168 deserves further attention, considering the potential molecular interaction of miR-168 with molecular cancer pathways. Further studies are needed to expand our knowledge of xeno-miRNA biogenesis in humans and particularly in gastric physiology.

## Figures and Tables

**Figure 1 diagnostics-13-02701-f001:**
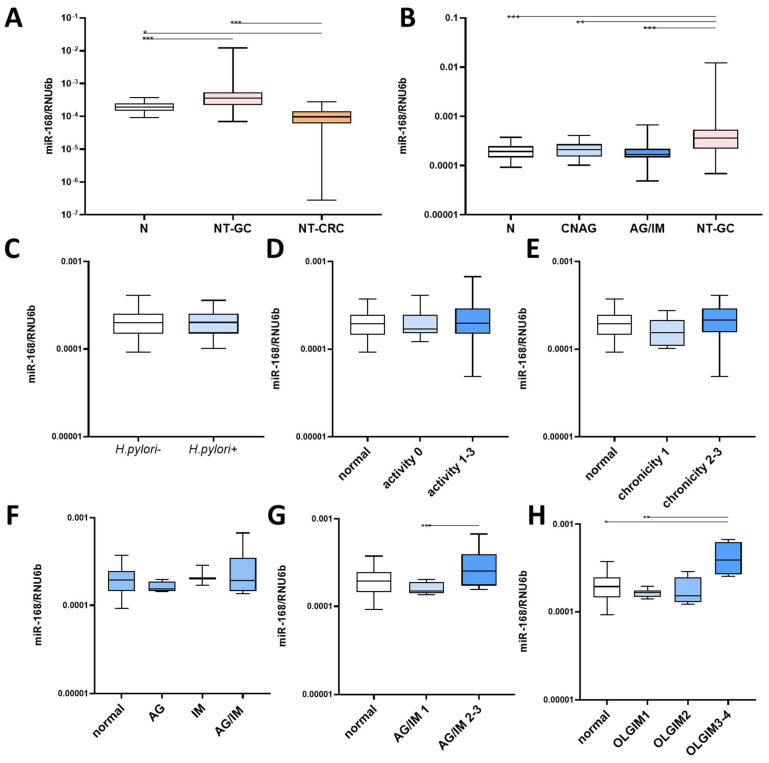
Concentration of miR-168 in relation to localization and inflammation pattern. (**A**) Comparison of miR-168 levels between normal (N), non-tumorous gastric mucosa (NT-GC) and non-tumorous colonic mucosa (NT-CRC); (**B**) miR-168 levels dependent on the type of gastritis: N, chronic non-atrophic gastritis (CNAG), atrophic gastritis (AG)/intestinal metaplasia (IM), non-tumour gastric cancer (NT-GC); (**C**) comparison of miR-168 between subjects with and without *H. pylori* infection (AG and IM excluded). miR-168 levels in the mucosa in relation to (**D**) activity, (**E**) chronicity, (**F**) presence of AG or IM, (**G**) degree of preneoplastic changes, and (**H**) OLGIM stages. All samples were normalized to RNU6b. Mann–Whitney U test or Kruskal–Wallis test with appropriate Dunn’s test was used for data analysis. Data presented as 2^ΔCT^ normalised to RNU6b. * *p* < 0.05; ** *p* < 0.01; *** *p* < 0.001.

**Figure 2 diagnostics-13-02701-f002:**
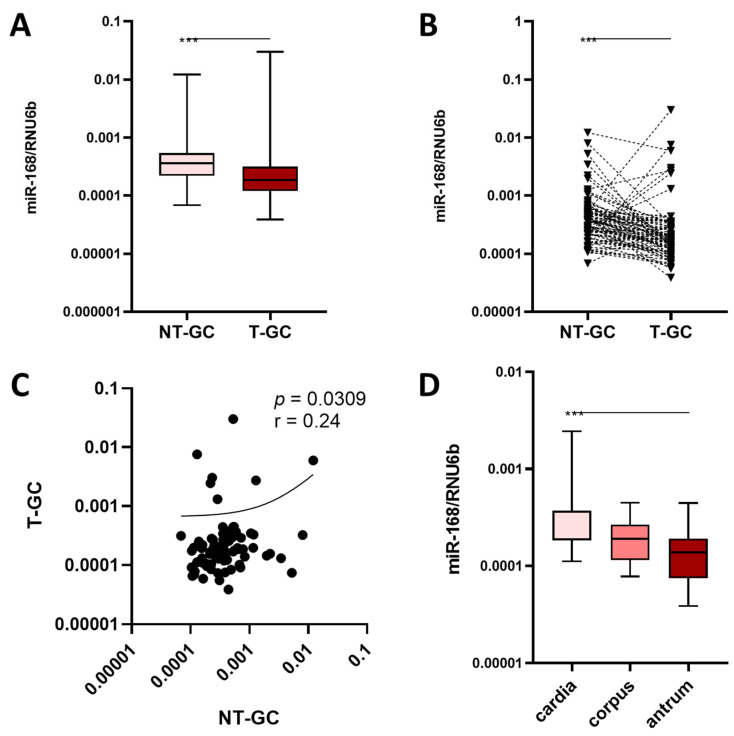
Analysis of miR-168 in GC patients. (**A**) Unpaired and (**B**) paired analysis of the miR-168 concentration between NT-GC and T-GC. in tumour localization between NT-GC and T-GC (gastric cancer tumour). (**C**) Correlation analysis between miR-168 levels in NT-GC and T-GC. (**D**) Localization of the GC tumour and miR-168 concentration. Mann–Whitney U-test was used for unpaired and Wilcoxon test for paired analyses. Spearman’s test was used for correlation analysis. Data presented as 2^ΔCT^ normalised to RNU6b. *** *p* < 0.001.

**Figure 3 diagnostics-13-02701-f003:**
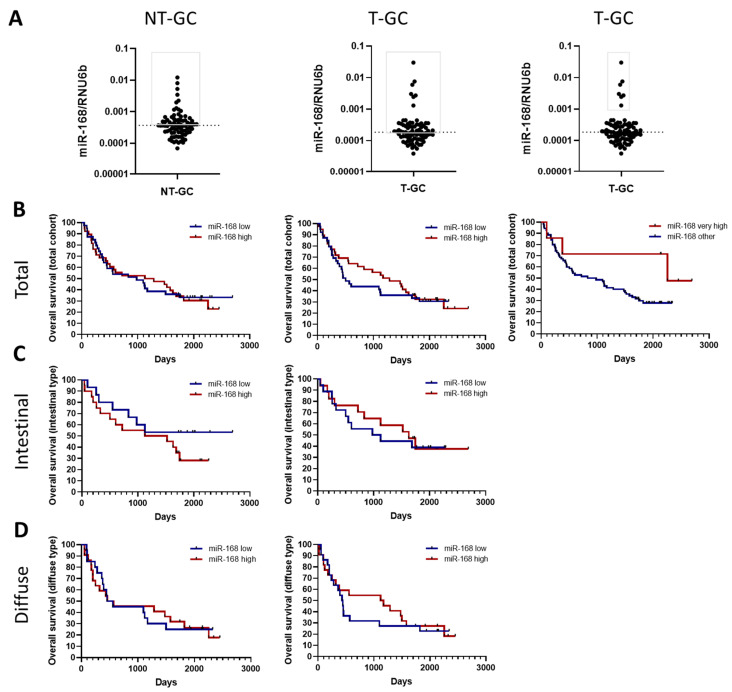
miR-168 and overall survival in GC patients. (**A**) Distribution of the miR-168 concentration in NT-GC, T-GC. The grey box refers to the definition of the low or high levels. (**B**) Overall survival analysis of all included in the study samples. (**C**) Overall survival analysis in a subgroup of patients with Lauren’s intestinal type of GC. (**D**) Overall survival analysis in a subgroup of patients with Lauren’s diffuse type of GC. Data presented as 2^ΔCT^ normalised to RNU6b.

**Figure 4 diagnostics-13-02701-f004:**
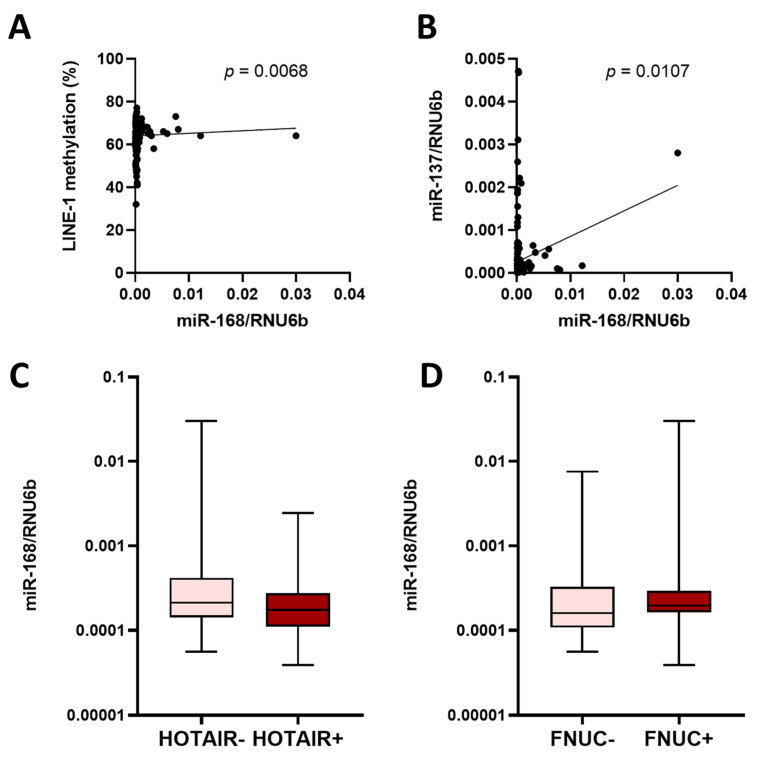
miR-168 levels and molecular characteristics of GC patients. (**A**) Correlation analysis between LINE-1 methylation and miR-168 level. (**B**) Correlation analysis between miR-137 CpG island promoter methylation and miR-168 level. miR-168 levels in tumours with and without (**C**) HOTAIR or (**D**) *F. nucleatum* positivity (FNUC). Data presented as 2^ΔCT^ normalised to RNU6b.

**Figure 5 diagnostics-13-02701-f005:**
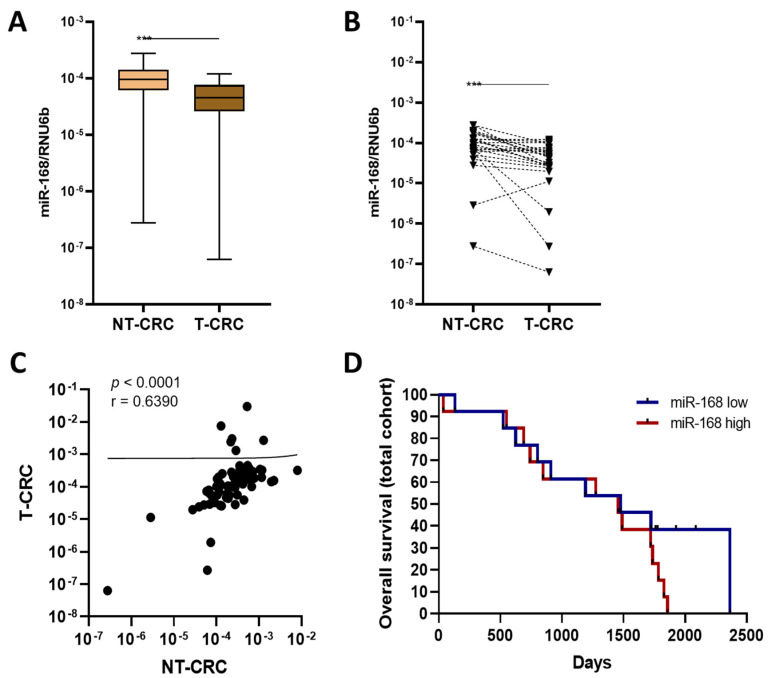
Analysis of miR-168 in CRC patients. (**A**) Unpaired and (**B**) paired analysis of miR-168 levels between NT-CRC and T-CRC. (**C**) Correlation analysis between miR-168 levels in NT-CRC and T-CRC. (**D**) Overall survival analysis of all samples included in the study. Data presented as 2^ΔCT^ normalised to RNU6b. *** *p* < 0.001.

**Figure 6 diagnostics-13-02701-f006:**
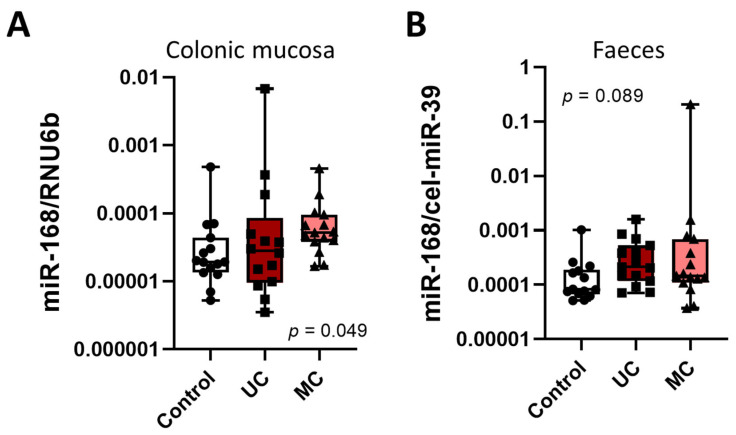
Analysis of miR-168 in IBD patients. miR-168 levels in colonic mucosa (**A**) and faeces (**B**) of controls and patients with active ulcerative colitis (UC) and Crohn’s disease (CD). Data presented as 2^ΔCT^ normalised to RNU6b for colon mucosa and to spiked cel-miR-39 for faeces. From a total of n = 15 in each cohort, one dataset from UC (mucosa) and one from controls (faeces) were excluded due to issues with normalization.

**Table 1 diagnostics-13-02701-t001:** Clinicopathological characteristics of gastric cancer patients in relation to miR-168 level.

	All		miR-168-Low		miR-168-High		*p*
	n = 81	%	n = 40	%	n = 41	%	
**Age**	65.85	±11.58	64.15	±10.76	67.50	±12.37	0.2 *
**Gender**							0.37
male	47	58%	21	52.5%	26	63%	
female	34	42%	19	47.5%	15	37%	
**Tumour localization**							0.04
cardia	8	10%	2	5%	6	15%	
corpus	45	56%	19	47.5%	26	63%	
antrum	28	34%	19	47.5%	9	22%	
**UICC**							0.33
I	16	20%	5	12.5%	11	26.8%	
II	21	26%	11	27.5%	10	24.4%	
III	36	44%	23	57.5%	13	31.7%	
IV	8	10%	1	2.5%	7	17.1%	
**T**							0.3
1 + 2	18	22%	6	15%	12	29%	
3	36	45%	19	47.5%	17	42%	
4	27	33%	15	37.5%	12	29%	
**N**							0.22
0	29	36%	11	27.59%	19	46.5%	
1	15	19%	8	20%	7	17%	
2	13	16%	6	15%	7	17%	
3	23	28%	15	37.5%	8	19.5%	
Unknown	1	1%			1	2.4%	
**M**							0.06
0	73	90%	39	97.5%	34	83%	
1	8	10%	1	2.5%	7	17%	
**Grading**							0.15
1	3	4%	0	0%	3	7%	
2	29	36%	13	32.5%	16	39%	
3	49	60%	27	67.5%	22	54%	
**Lauren Classification**							0.14
Diffuse Type	44	54%	23	57.5%	21	51%	
Intestinal Type	26	32%	12	30%	14	34%	
Mixed Type	7	9%	5	12.5%	2	5%	
Unknown	4	5%			4	10%	
*H. pylori*							0.02
Negative	8	10%	1	2.5%	7	17%	
Positive	17	21%	6	15%	11	27%	
Unknown	56	69%	33	82.5%	23	56%	

Legend: miR-168 low and high were defined by the median of 2^ΔCT^ of miR-168 normalised to RNU6b. * unpaired *t*-test. UICC: Union for International Cancer Control; T: primary tumour stage; N: lymph node metastasis staging; M: metastasis staging.

## Data Availability

The data are available from the corresponding author upon reasonable request.

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
