# Peer review of "The Translational Impact of Plant-Derived Xeno-miRNA miR-168 in Gastrointestinal Cancers and Preneoplastic Conditions"

_diagnostics, 2023, doi:10.3390/diagnostics13162701_

Round 1

Reviewer 1 Report

Summary

                Plant-derived microRNAs acquired through diet have an unknown contribution to carcinogenesis. One of the most abundant exogenous miRNAs is miR-168. In order to better understand the contribution of miR-168 to cancers of the GI tract, the authors of this study analyzed the expression of this miRNA in stomach tissues from patients in different phases of disease leading toward gastric cancer (GC) as well as colorectal cancer (CRC) tissues. It was found that miR-168 expression was higher in GC as opposed to CRC. There were also higher levels of miR-168 in non-tumor tissue compared to tumors. At the same time, expression of miR-168 was not associated with preneoplastic disease stage or H. pylori infection. Interestingly, patients with the highest expression of miR-168 in GC tumors had a somewhat improved prognosis. The findings of this study indicate that further research into miR-168 could be valuable.

Issues

1.       I assume that “2ΔΔCT” is supposed to be 2^(-ΔΔCt), which is the standard equation for calculating fold change. If “2ΔΔCT” is not a mistake, however, a citation for this calculation method needs to be provided.

2.       The Y-axis labels of many graphs throughout the paper are not sufficiently detailed. For example, the axis label “miR-168/RNU6b” does not indicate which numbers were used to calculate this ratio. Was it the ΔCt? The ΔΔCt? Etc.

3.       The generally agreed-upon standard for graphing qPCR data is to show the results in terms of fold change in expression over the baseline or control. This is an intuitive and informative way of looking at the data. Apart from previously published papers by the authors of this study, I have not encountered the practice of displaying qPCR data as a ratio of the gene of interest to the reference gene. Since normalization to the reference gene is built into the calculation of ΔCt or ΔΔCt (depending on the method), there is no further need to relate the expression of the gene of interest to that of the reference gene.

4.       The raw Ct values in Figure 6 parts A and C are unnecessary.

5.       The statement in the first section of 3.2, “there was a lower level of miR-168 in NT-GC compared to T-GC,” contradicts what is shown in Figure 2A and what is concluded in the second paragraph of the Discussion.

6.       Epstein-Barr virus (EBV) is known to be a factor in the development of some gastric cancers, possibly even having synergistic effects with H. pylori [1 and 2]. Furthermore, this virus has been shown to manipulate the expression/activity of host miRNAs [3]. EBV is potentially a major confounding factor in the miR-168 expression data. The EBV status of the tissue samples should be determined, if possible.

7.       The relevance of the molecular features described in section 3.4 (LINE-1 DNA methylation, HOTAIR expression, etc.) is not explained and no papers are cited.

Minor Issues

8.       The term “post-translational” in the second line of the introduction should be “post-transcriptional”.

9.       Sections 2.3 and 2.5 have sentences which are written twice.

10.   “Preneoplastic” is misspelled on multiple occasions.

11.   H. pylori is sometimes not italicized.

Citations

1.       Hirabayashi M, Georges D, Clifford GM, de Martel C. Estimating the Global Burden of Epstein-Barr Virus-Associated Gastric Cancer: A Systematic Review and Meta-Analysis. Clin Gastroenterol Hepatol. 2023 Apr;21(4):922-930.e21. doi: 10.1016/j.cgh.2022.07.042. Epub 2022 Aug 10. PMID: 35963539.

2.       Kashyap D, Rele S, Bagde PH, Saini V, Chatterjee D, Jain AK, Pandey RK, Jha HC. Comprehensive insight into altered host cell-signaling cascades upon Helicobacter pylori and Epstein-Barr virus infections in cancer. Arch Microbiol. 2023 Jun 13;205(7):262. doi: 10.1007/s00203-023-03598-6. PMID: 37310490.

3.       Ungerleider N, Bullard W, Kara M, Wang X, Roberts C, Renne R, Tibbetts S, Flemington EK. EBV miRNAs are potent effectors of tumor cell transcriptome remodeling in promoting immune escape. PLoS Pathog. 2021 May 6;17(5): e1009217. doi: 10.1371/journal.ppat.1009217. PMID: 33956915; PMCID: PMC8130916.

Author Response

We would like to thank the editor and reviewers for their valuable comments. We have addressed the comments point by point and hope to improve the quality of the work according to the suggestions.

Point by point response

Comments and Suggestions for Authors

Reviewer 1

Summary: Plant-derived microRNAs acquired through diet have an unknown contribution to carcinogenesis. One of the most abundant exogenous miRNAs is miR-168. In order to better understand the contribution of miR-168 to cancers of the GI tract, the authors of this study analyzed the expression of this miRNA in stomach tissues from patients in different phases of disease leading toward gastric cancer (GC) as well as colorectal cancer (CRC) tissues. It was found that miR-168 expression was higher in GC as opposed to CRC. There were also higher levels of miR-168 in non-tumor tissue compared to tumors. At the same time, expression of miR-168 was not associated with preneoplastic disease stage or H. pylori infection. Interestingly, patients with the highest expression of miR-168 in GC tumors had a somewhat improved prognosis. The findings of this study indicate that further research into miR-168 could be valuable.

Issues

  1. I assume that “2ΔΔCT” is supposed to be 2^(-ΔΔCt), which is the standard equation for calculating fold change. If “2ΔΔCT” is not a mistake, however, a citation for this calculation method needs to be provided.

Thanks for pointing that out. Indeed, throughout the paper we used either raw CT values or 2ΔCT. We have corrected the description..

  1. The Y-axis labels of many graphs throughout the paper are not sufficiently detailed. For example, the axis label “miR-168/RNU6b” does not indicate which numbers were used to calculate this ratio. Was it the ΔCt? The ΔΔCt? Etc.

As suggested by the reviewer, we have revised the legends and/or figures to make the data more understandable. As pointed out, in almost all cases the data refer to 2ΔCT values.

  1. The generally agreed-upon standard for graphing qPCR data is to show the results in terms of fold change in expression over the baseline or control. This is an intuitive and informative way of looking at the data. Apart from previously published papers by the authors of this study, I have not encountered the practice of displaying qPCR data as a ratio of the gene of interest to the reference gene. Since normalization to the reference gene is built into the calculation of ΔCt or ΔΔCt (depending on the method), there is no further need to relate the expression of the gene of interest to that of the reference gene.

We agree with the reviewer that the 2ΔCT or ΔCT values provide sufficient information to describe the values. The data presented in the paper are presented as 2ΔCT with the exception of 2 Figures 6A and 6C. We understand that the recommendation is to delete the figures as stated in the next comment. Figures 6A and 6C with raw values have been deleted at the request of the reviewers.

  1. The raw Ct values in Figure 6 parts A and C are unnecessary.

The figure 6A and 6C with raw values were deleted as recommended.

  1. The statement in the first section of 3.2, “there was a lower level of miR-168 in NT-GC compared to T-GC,” contradicts what is shown in Figure 2A and what is concluded in the second paragraph of the Discussion.

We apologize for the typos in T vs. NT. The statement, as correctly pointed out by the reviewer, has been corrected.

  1. Epstein-Barr virus (EBV) is known to be a factor in the development of some gastric cancers, possibly even having synergistic effects with  pylori [1 and 2]. Furthermore, this virus has been shown to manipulate the expression/activity of host miRNAs [3]. EBV is potentially a major confounding factor in the miR-168 expression data. The EBV status of the tissue samples should be determined, if possible.

As the reviewer correctly points out, there are several potential molecular factors associated with carcinogenesis. Among these, and likely to be highly relevant to prognosis and treatment, is EBV. In our study, we included 81 paired samples from GC patients, which would have resulted in an EBV-positive tumour rate of 4-5 patients based on current estimates. Due to this estimation, we did not perform these analyses initially, but as suggested by the reviewer, we hope to address these issues in the future with a larger cohort to avoid sample size bias. Nevertheless, we have updated the references and discussion on this topic.

  1. The relevance of the molecular features described in section 3.4 (LINE-1 DNA methylation, HOTAIR expression, etc.) is not explained and no papers are cited.

We appreciate the comment. As suggested by the reviewer, we have revised the Results section to refer to the previous work related to the molecular features analysed (3.4).  

Minor Issues

  1. The term “post-translational” in the second line of the introduction should be “post-transcriptional”.

Corrected.

  1. Sections 2.3 and 2.5 have sentences which are written twice.

Corrected.

  1. “Preneoplastic” is misspelled on multiple occasions.

Corrected.

  1. H. pyloriis sometimes not italicized.

Corrected.

Citations

  1. Hirabayashi M, Georges D, Clifford GM, de Martel C. Estimating the Global Burden of Epstein-Barr Virus-Associated Gastric Cancer: A Systematic Review and Meta-Analysis. Clin Gastroenterol Hepatol. 2023 Apr;21(4):922-930.e21. doi: 10.1016/j.cgh.2022.07.042. Epub 2022 Aug 10. PMID: 35963539.
  2. Kashyap D, Rele S, Bagde PH, Saini V, Chatterjee D, Jain AK, Pandey RK, Jha HC. Comprehensive insight into altered host cell-signaling cascades upon Helicobacter pyloriand Epstein-Barr virus infections in cancer. Arch Microbiol. 2023 Jun 13;205(7):262. doi: 10.1007/s00203-023-03598-6. PMID: 37310490.
  3. Ungerleider N, Bullard W, Kara M, Wang X, Roberts C, Renne R, Tibbetts S, Flemington EK. EBV miRNAs are potent effectors of tumor cell transcriptome remodeling in promoting immune escape. PLoS Pathog. 2021 May 6;17(5): e1009217. doi: 10.1371/journal.ppat.1009217. PMID: 33956915; PMCID: PMC8130916.

The suggested references have been included in the work. Thank you very much.

Reviewer 2 Report

The manuscript on "Translational impact of the plant-derived xeno-miRNA miR-168 in gastrointestinal cancers and preneoplastic conditions" by Link et al. is an interesting study which will help in understanding the role of xeno-miRNA168 in gastric and colorectal cancers. Overall, it is well-written but there are a few suggestion/comments to improve the manuscript. 

1- There are a few language mistakes so please go through the manuscript one more time to avoid any language errors. 

2- Be consistent in using the word "miRNA" and "miRNAs". 

3- Second paragraph of the introduction starts with "recently" which is not true because the very first time miRNA was detected in plants back in 2002. 

4-Add the reference of "Those miRNAs that have been identified in other species or even across kingdoms are called xeno-miRNAs".

5- In methodology, DNA and RNA Isolation is mentioned as heading, however, no details of DNA isolation are included. 

6- The discussion section should be improved with more references from the literature. 

It can be improved.

Author Response

We would like to thank the editor and reviewers for their valuable comments. We have addressed the comments point by point and hope to improve the quality of the work according to the suggestions.

Point by point response

Reviewer 2

Comments and Suggestions for Authors

The manuscript on "Translational impact of the plant-derived xeno-miRNA miR-168 in gastrointestinal cancers and preneoplastic conditions" by Link et al. is an interesting study which will help in understanding the role of xeno-miRNA168 in gastric and colorectal cancers. Overall, it is well-written but there are a few suggestion/comments to improve the manuscript.

1- There are a few language mistakes so please go through the manuscript one more time to avoid any language errors.

Thank you for for pointing this out. We have double checked for typos and revised the text again.

2- Be consistent in using the word "miRNA" and "miRNAs".

The wording has been revised.

3- Second paragraph of the introduction starts with "recently" which is not true because the very first time miRNA was detected in plants back in 2002.

We apologise for the misleading statement. In fact, the first paper identifying microRNA in plants dates back to 2002 (the reference is included). However, the identification of plant microRNA in humans is still in its infancy. We have revised the wording.

4-Add the reference of "Those miRNAs that have been identified in other species or even across kingdoms are called xeno-miRNAs".

We've included the first reference we know of to Xeno-miRNAs.

5- In methodology, DNA and RNA Isolation is mentioned as heading, however, no details of DNA isolation are included.

The information on DNA extraction has been referenced and is now included in the methods as recommended by the reviewer.

6- The discussion section should be improved with more references from the literature.

As requested by the reviewer, we have expanded the Discussion section to include the new functional data related to xenomiRNAs.

Round 2

Reviewer 1 Report

Although I still find it very unusual and less than ideal, after clarification I am willing to accept the method of calculation used on the qPCR data. Thank you for including the issue of EBV-related gastric cancers in the discussion.